# Detection, Identification, and Diffusion of Yeasts Responsible for Structural Defects in Provolone Valpadana PDO Cheese Using Multiple Research Techniques

**DOI:** 10.3390/foods15010129

**Published:** 2026-01-01

**Authors:** Miriam Zago, Barbara Bonvini, Lia Rossetti, Milena Povolo, Luca Ballasina, Vittorio Emanuele Pisani, Flavio Tidona, Giorgio Giraffa

**Affiliations:** 1Council for Agricultural Research and Economics, Research Centre for Animal Production and Aquaculture (CREA-ZA), 26900 Lodi, Italy; barbara.bonvini@crea.gov.it (B.B.); lia.rossetti@crea.gov.it (L.R.); milena.povolo@crea.gov.it (M.P.); flavio.tidona@crea.gov.it (F.T.); giorgio.giraffa@crea.gov.it (G.G.); 2Consorzio Tutela Provolone Valpadana, 26100 Cremona, Italy; assistenza@provolonevalpadana.it (L.B.); direzione@provolonevalpadana.it (V.E.P.)

**Keywords:** cheese spoilage, gas-forming microbiota, dairy yeasts, microbial identification, metabarcoding analysis, molecular typing, Provolone Valpadana PDO cheese

## Abstract

The aim of this work was to identify the microbial agent(s) responsible for a structural defect in Provolone Valpadana, Protected Designation of Origin (PDO) cheese, and to establish their spread along the production line. Repeated sampling of defective cheeses and analyses of processing intermediates following two inspections at the cheese factory identified yeasts as the main causative agents. Microbiological analysis highlighted an almost constant presence of yeasts, which dominate over the other microbial groups. Forty yeast isolates from defective cheeses were identified by sequencing the D1/D2 region of the 26S rRNA gene. *Saccharomyces cerevisiae, Kluyveromyces marxianus*, and *Debaryomyces hansenii* dominated in all sampled cheeses, followed by *D. tyrocola*, *Pichia kudriavzevii*, and *Torulaspora delbrueckii*. Yeast and bacterial metabarcoding on three cheeses with a yeast count > log 4.0 CFU/mL indicated *D. hansenii* as the dominant yeast taxon and confirmed the absence of gas-producing bacterial taxa. RAPD-PCR analysis suggested the presence of yeast biofilms in the dairy environment or along the production line, as confirmed by the repeated isolation of specific genotypes of *S. cerevisiae*, *K. marxianus*, and *D. hansenii* in different defective cheeses sampled between April and August 2023, as well as in samples taken following two inspections at the production site, during cheese processing and ripening.

## 1. Introduction

Provolone belongs to the group of semi-hard, cooked, and curd-stretched cheeses produced with whole cow’s milk. Today, the term “Provolone” is reserved for three different types of cheeses, one produced with the generic name Provolone and two Protected Designation of Origin (PDO) varieties: the more recent Provolone del Monaco and the historic Provolone Valpadana (PV). To satisfy different consumer preferences, this latter is produced in two different technological variants (“sweet” or mild and “spicy”) [1]. Provolone Valpadana has a reduced moisture content (less than 46%) because of the processing method and ripening, which may last 12–24 months for the large-sized spicy variety. The key points of PV cheesemaking are the following: pasteurization of milk (only to produce the sweet variety); curd cooking at a temperature generally between 48 and 52 °C; lactic fermentation, carried out by a natural, thermophilic whey starter culture (NWC) in curd blocks (without molds), until reaching pH values close to 4.7–5.2 (defined as ‘curd maturation’); curd stretching, carried out with hot water at a temperature ≥60 °C, followed by shaping and cooling in water; and salting in brine, tying and, ripening, which complete the production process. Sizes may vary from 6 to 30 kg, while typical shapes are defined as “salami”, the most common one, or “pear”, the most distinguished one (Appendix A). Ripening time depends on the cheese size, with a minimum with a minimum of 30 and of 60 d for the sweet and spicy variants, respectively. This latter is characterized by lipolytic maturation, obtained using rennet paste. The cheese paste is compact, although a limited presence of small- and/or medium-sized eyes is tolerated and, especially in the ‘spicy’ variant, even typical [2]. The peculiar production and maturation methods and the chemical–physical characteristics of stretched curd cheeses, such as PV and similar cheeses, are a typical example of the so-called hurdle approach, in which different physical–chemical factors, acting synergistically, contribute to reducing or limiting undesired microbial proliferation during shelf life.

Despite its commercial importance and relevance in the Italian dairy tradition, studies on the microbiology of Provolone cheese are surprisingly limited. No cases of the presence of pathogenic microorganisms have been described so far in Provolone cheese, whereas deterioration due to spoilage bacteria has sometimes been reported in similar cheeses, such as Grottone, a pasta filata hard cheese produced in Campania from cow milk [3,4,5,6]. However, most cheese products remain susceptible to growth for a variety of fungal species that can thrive under low pH and low a_w_ conditions, or, beyond fungi, cheese spoilage may occur by bacterial contamination that, differently from fungal spoilage, may take place in products with relatively high pH and a_w_ [7,8,9]. In “spicy”-type PV produced using milk stored at 6 and 12 °C for 60 h, no health risks or low hygiene indicators were found even when the raw milk had high microbial loads, highlighting the effectiveness of PV processing in reducing or eliminating contaminating microorganisms [1]. Non-starter lactic acid bacteria (NSLAB) belonging to thermophilic species such as *Streptococcus* (*S.*) *thermophilus*, *Lacticaseibacillus* (*Lcb.*) *rhamnosus*, and enterococci, which often reach levels of up to 108 CFU/g, can be found in Provolone cheese at the end of ripening [7,10,11].

Dairy products are the third food category in terms of the estimated total value of food losses. More than 12% of the dairy production (including cheeses) is wasted in Europe; notably, most waste and losses originate from processing, ripening, and/or consumption stages [12]. Microbial spoilage of cheeses is difficult to recognize and classify because it is generally caused by a variety of microorganisms and results from multiple and concomitant factors, linked to the quality of the raw material and the observance of process hygiene [8,9,13]. Cheese defects can be schematically grouped into modification of the structure and consistency of the paste (swelling, cracks, softening, chalky appearance); surface or crust spoilage (abnormal coloring, excessive mold proliferation); and biochemical irregularities during ripening (bitterness, lipolysis), leading to modification of taste, flavor, and color [8]. Here, we describe a defect in the PV (sweet variety) of microbial origin, consisting of an extended eye formation in the cheese due to gas accumulation during ripening. This study had two objectives: to identify the gas-forming microbiota responsible for this defect and to detect the process phases potentially involved in the contamination and/or development of gas-forming microbiota. In this regard, monitoring of the gas-forming microbial groups in different phases of the manufacturing process was carried out.

## 2. Materials and Methods

### 2.1. Cheese Sampling

Nine PV samples, labeled from A to I (Table 1) and produced between April and August 2023 in a factory recognized for the PV PDO manufacture, were collected when the defect was clearly manifested, i.e., after approx. 60–70 d of ripening. A 5 cm thick slice of cheese, representative of the entire diameter of the form, was cut, vacuum-packed, and sent to the laboratory of CREA-ZA for microbiological analysis. Since the defect was frequent but did not affect all the batches produced nor all the cheeses of an altered batch, cheeses were sectioned before sampling to identify those that, after visual inspection, presented extended micro-holes due to excessive gas accumulation. Additionally, on-site sampling was carried out during two separate inspections (dated 21 September 2023 and 6 February 2024) at the cheese factory to identify, along the production line, any critical process points potentially involved in the onset of the defect. The choice of these two dates was due to precise indications provided by the factory and related to changes in the quality of the milk to be processed. To this end, the following samples (10 g or mL) were taken: drained whey, part of which will be used to prepare the NWC for the next day’s processing (back slopping); NWC; curd before maturation; curd after maturation (before stretching); curd after stretching; and cheeses after 30, 60, and 90 d of ripening.


foods-15-00129-t001_Table 1Table 1Microbiological analysis of defective Provolone Valpadana (PV) cheeses and of samples taken from cheese production lines during two on-site inspections (21 September 2023 and 6 February 2024). Counts are expressed as log10 CFU/g (or/mL for samples of drained whey). SD is related to the average of two replicates.Different Defective Cheese Batches




Sample nameColiformsHeterofermentative LAB*Leuconostoc* spp.Spore-forming bacteriaYeastsMoldsPropionibacteriaA<0.70<0.70<0.702.49 ± 0.013.23 ± 0.05<1.70<0.70B<0.70<0.703.39 ± 0.39<1.704.08 ± 0.03<1.70<0.70C<0.70<0.70<0.702.80 ± 0.023.75 ± 0.15<1.70<0.70D<0.70<0.70<2.703.33 ± 0.033.45 ± 0.152.27 ± 0.572.78 ± 0.07E<0.70<0.70<2.702.63 ± 0.153.19 ± 0.192.57 ± 0.57<0.70F<0.70<0.70<2.702.91 ± 0.134.06 ± 0.02<1.70<0.70G<0.70<0.70<2.704.31 ± 0.033.72 ± 0.12<1.70<0.70H<0.70<0.70<2.70<1.704.76 ± 0.01<1.70<0.70I<0.70<0.70<2.70<1.703.77 ± 0.02<1.70<0.70On-site inspection of 21 September 2023      SamplesColiformsHeterofermentative LAB*Leuconostoc* spp.Spore-forming bacteriaYeastsMoldsPropionibacteriadrained whey1.2 ± 0.05<0.50<0.50<1.70<1.70<1.702.23 ± 0.05whey starter<0.50<0.50<0.50<0.70<1.70<1.70<0.50curd before maturation<0.70<0.70<0.70<1.701.98 ± 0.02<1.70<0.70curd after maturation<0.70<0.70<0.70<1.702.89 ± 0.11<1.701.54 ± 0.06curd after stretching<0.70<0.70<0.70<1.703.40 ± 0.03<1.702.02 ± 0.02PV cheese 30 days<0.70<0.70<1.70<0.704.28 ± 0.01<1.70<1.70PV cheese 60 days<0.70<0.70<1.70<1.702.30 ± 0.30<1.70<1.70PV cheese 90 days<0.70<0.70<1.70<1.70<1.70<1.701.48 ± 0.04On-site inspection of 6 February 2024      SamplesColiformsHeterofermentative LAB*Leuconostoc* spp.Spore-forming bacteriaYeastsMoldsPropionibacteriadrained whey3.24 ± 0.06<0.501.04 ± 0.04<0.502.74 ± 0.05<0.50<0.50whey starter<0.5<0.50<0.50<0.502.58 ± 0.04<0.50<0.50curd before maturation0.92 ± 0.08<0.70<0.70<0.702.02 ± 0.02<0.701.10 ± 0.20curd after maturation<0.70<0.70<0.70<0.701.78 ± 0.18<0.700.89 ± 0.11curd after stretching<0.70<0.70<0.70<0.70<0.70<0.700.89 ± 0.11PV cheese 30 days<0.70<0.70<1.70<1.704.81 ± 0.01<0.70<1.70PV cheese 60 days<0.70<0.70<1.70<0.702.66 ± 0.01<0.70<0.70PV cheese 90 days<0.70<0.70<1.701.59 ± 0.113.27 ± 0.01<0.70<0.70


### 2.2. Microbiological Analysis

Coliforms and *Escherichia coli* were enumerated on ChromID Coli agar (Biomerieux, Bagno a Ripoli, Italy) at 37 °C for 24 h; heterofermentative LAB were enumerated on MRS broth (Oxoid, Basingstoke, UK) with Durham tubes at 37 °C for 48 h; *Leuconostoc* spp. were enumerated on MSE agar (Biolife, Milan, Italy) at 21 °C for 4 d; yeasts and molds were enumerated on Yeast Glucose Chloramphenicol (YGC, Biolife, Milan, Italy) agar; propionibacteria were enumerated on Pal Propiobac agar medium (Laboratoires Standa, Caen, France) at 30 °C for 6 d under anaerobic conditions; and, finally, butyric clostridia spores (BCS) were counted by the Most Probable Number (MPN) technique according to the method described previously [14]. The samples were analyzed in duplicate.

### 2.3. Yeast Isolation and Purification

For each cheese sample, three to five colonies were randomly isolated from YGC agar plates (Biolife) and grown on the corresponding broth medium at 25 °C for 3–5 d three consecutive times. A total of 60 isolates were preliminarily examined by optical microscopy to check yeast morphology and then stored at −80 °C with 15% glycerol as cryoprotectant.

### 2.4. DNA Extraction and Genotyping of the Yeast Isolates

The yeast isolates were cultivated on YGC broth (Oxoid) at 25 °C for 24–48 h. Total DNA was extracted from the purified isolates by the QIAcube HT automated station (Qiagen, Milan, Italy) using QIAamp 96 QIAcube HT kit (Qiagen) and quantified fluorometrically (Qubit™, Life Technologies, Monza, Italy). The isolates were typed by RAPD-PCR fingerprinting using primer M13 (5′-GAGGGTGCGGTTCT-3′) according to Rossetti and Giraffa [15]. PCR products were separated by capillary electrophoresis with QIAxcel (Qiagen), using QIAxcel DNA Screening Gel Cartridge (Qiagen) according to the procedure described previously [14]. Cluster analysis of RAPD-PCR profiles was carried out by BioNumerics™ software (version 7.6, Applied Maths, Belgium), applying the Pearson correlation coefficient and the Ward method as clustering algorithm. Dendrograms were set at a similarity value of 75%, which is the internal reproducibility level of the RAPD-PCR [15].

### 2.5. Isolate Identification

Sequencing of the D1/D2 region of the 26S rRNA gene was carried out on isolates belonging to different RAPD genotypes using primers NL-1 (5′-GCA TAT CAA TAA GCG GAG GAA AAG-3′) and NL-4 (5′-GGT CCG TGT TTC AAG ACG G-3′) described by Geronikou et al. [16]. PCR reactions were carried out in a 30 µL volume mixture containing 2.5 U of Taq DNA Polymerase Gold, 0.5 µM of each primer, 1.5 µM MgCl2, 0.2 mM dNTPs, 1X buffer Gold 1, and 2 µL of the total DNA from yeast. The PCR reaction was carried out on a ProFlex PCR System (Thermofisher, Milan, Italy) under the following conditions: initial denaturation for 10 min at 95 °C, followed by 30 cycles at 95 °C for 60 s, 53 °C for 30 s, and 72 °C for 60 s, and the final elongation step at 72 °C for 7 min. Sequencing reactions using primers NL-1 and NL-4 were carried out on a SeqStudio genetic analyzer using a SeqStudio Cartridge v2 (Thermofisher) with a 5 s injection time, dye set of Z_BigDye Terminator BDT v3.1, and a medium seq module setting. The species assignment was performed through BlastN (www.ncbi.nlm.nih.gov/BLAST, accessed on 23 June 2024) alignment of the obtained sequences (sized approx. 600 bp) with the 26S rRNA gene sequences of yeasts available from the EMBL database. Alignment of the 26S rRNA gene sequences was carried out using Clustal Omega online software (https://www.ebi.ac.uk/Tools/msa/clustalo/, accessed on 29 October2024).

### 2.6. Metagenomic Analysis

A 16S and ITS metabarcoding analysis was carried out on total DNA extracted from three single Provolone cheese samples, with a count of yeast >log 4 CFU/mL. Bacterial DNA was extracted following the protocol as described by Zago et al. [17]. Yeast DNA was extracted with the same protocol as bacterial DNA using lyticase (50 U/sample) (Merck, Milan, Italy) instead of lysozyme. The 16S Ion Metagenomics Kit protocol (Thermo Fisher Scientific, Segrate, Italy) was used for 16S library preparation, ITS library was prepared using primers for the ITS and LSU regions (given by Thermo Fisher Scientific) and TaqMan Environmental Mastermix (Thermo Fisher Scientific) following the 16S Ion Metagenomics Kit protocol. The amplified hypervariable regions were pooled together and diluted to obtain a quantity of 50 ng. Ion Xpress Plus Fragment Library Kit (Thermo Fisher Scientific) was used to ligate sample barcodes and amplify the library. Sequencing run on a 510 Ion chip was carried out on Ion Gene Studio S5 plus (Thermo Fisher Scientific) using 850 flows for 450-bases-read sequencing. Data analysis was performed using R software version 4.4.1 (http://www.r-project.org/index.html, accessed on 31 October 2024). OTUs < 10 reads were removed. On the resulting OTU tables, the relative abundance for each OTU across the samples was calculated, and only “dominant” OTUs were discriminated as >1% of relative abundance. Taxonomic analysis of the bacterial communities was performed and visualized by using “reshape2” and “ggplot2” packages, according to Zago et al. [17].

### 2.7. Chemical Analysis

pH, sugars, and organic acids were determined in the curd during acidification and in the cheeses, according to Bouzas et al. [18]. Volatile alcohols were evaluated by SPME/GC/MS. A DVB/CAR/PDMS, 50/30 μm, 2 cm long fiber (Supelco, Bellefonte, PA, USA) was used to collect volatile fractions by SPME. Five grams of grated cheese was weighed in a 20 mL crimp-top vial and sealed with an aluminum cap provided with a pierceable septum (23 × 75 mm, Varian, Palo Alto, CA, USA). The sample was allowed to equilibrate to 45 °C in a thermostatic bath for 5 min without agitation, and the fiber was exposed to the headspace for 30 min. The gas chromatographic analysis was carried out with a CP-WAX 52CB capillary column (Varian; 60 m long, 0.32 mm i.d., 0.5 μm film thickness). An Agilent 7890 A gas chromatograph coupled with an Agilent 5975C mass spectrometer was used (Palo Alto, CA, USA), adopting the conditions reported by Lazzi et al. [19]. The identification of volatile compounds was performed with the following criteria: comparison with the mass spectra of the W8 N08 library (John Wiley and Sons, Inc., New York, NY, USA), injection of authentic standards analyzed under the same GC-MS conditions, and calculation of retention indices (RI) followed by comparison with those obtained from both authentic standards and literature. Values were expressed as area units ×10^−6^.

## 3. Results

### 3.1. Description of the Defect

The appearance of the defect in sweet PV cheese, aged approximately 60–70 days, is shown (Appendix A). To note, ripened PV cheese does not normally present gas holes (Appendix A). The defect occurred on cheeses produced in spring 2023 and involved approx. 10% cheese per production batch. The characteristics of the spoiled cheeses were as follows: small-/medium-sized holes, irregularly shaped, across the whole cheese (from the under-rind to the cheese core); openings in the peripheral areas of the cheese; and the absence of cheese blowing or unpleasant odors and flavors. The characteristics of the defect, suggestive of gas accumulation, addressed the search for the spoilage agent towards gas-producing microbial groups, possibly outgrowing during cheese production and ripening.

### 3.2. Microbiological Analysis

Table 1 reports the outcomes of a preliminary series of microbiological analyses on various batches of sweet PV ripened for 60–70 d and affected by the defect, with the aim of postulating which groups of gas-forming microorganisms could be causative of the anomalies described above. Results highlighted an almost constant presence of yeasts, which dominate over the other microbial groups (coliforms, heterofermentative LAB, anaerobic spore-forming bacteria, and propionibacteria). Notably, spore-forming bacteria dominated over the whole microbial community in D and G defective PV samples (reaching >log 3.0 CFU g^−1^; Table 1). Probably, anaerobic sporeformers contributed, together with yeasts, to the onset of the defect in these two samples. Following this step, a monitoring of the sweet PV production line was carried out to investigate the phases of the manufacturing process potentially involved in the development of the gas-forming microbiota. The on-site inspections at the dairy plant, which were repeated twice on two separate dates, showed the following: (i) a significant increase in yeasts during curd acidification (maturation), especially on the first on-site inspection; (ii) a poor effectiveness of curd stretching, carried out in this dairy with hot water at 58 °C, in reducing the yeast population; (iii) an overgrowth of yeasts up to 30 days of ripening, where they reached their maximum level (4.80 log CFU g^−1^), leading to defective cheeses at 60–70 d of ripening (Figure 1); (iv) a rapid mortality of yeast cells between 30 and 60 d; (v) and a consistent number of yeast survivors at 90 d ripening in cheeses sampled on 6 February 2024. The levels of the other microbial groups investigated remained low or undetectable during both cheese production and ripening (Table 1). Numerous studies demonstrate that aerosols, in the form of droplets or dust, can be easily dispersed by air flows, introducing a major route of transmission of microbial contaminants, especially ubiquitous yeasts, in the food environment [20].

**Figure 1 foods-15-00129-f001:**
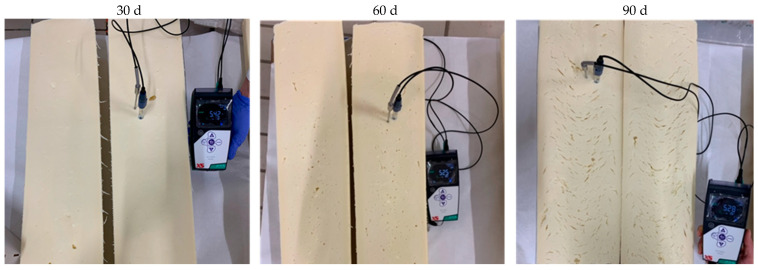
Provolone Valpadana cheeses sampled after 30, 60, and 90 days of ripening during the on-site inspections.

### 3.3. Yeast Isolation, RAPD Typing, and Molecular Identification

With the aim of better focusing on the yeast species involved in the onset of the defect, dominant isolates, i.e., those from the most diluted cheese samples for a total of 60 colonies, were initially taken from the YGC agar plates used for yeast enumeration in different batches of sweet PV and in samples taken during the two cheese plant surveys. After growth on the corresponding broth medium and plate purification, 40 out of the 60 isolates were selected: 10 isolates came from the 9 batches of defective PV cheeses, and 30 isolates came from cheese samples collected on 21 September 2023 (10 isolates) and 26 February 2024 (20 isolates) during ripening. Cultures were examined by phase contrast microscopy to verify typical yeast cell morphology. Total DNA was extracted, and isolates were typed by RAPD-PCR, which split them into different genotypic groups; representative profiles of each group were then identified by sequencing the D1/D2 region of the 26S rRNA gene. Isolate identification revealed the presence of fermentative yeast species, typically present in cheeses, such as *Saccharomyces cerevisiae*, *Kluyveromyces marxianus*, and *Debaryomyces hansenii*, *D. tyrocola*, *Pichia kudriavzevii*, and *Torulaspora delbrueckii* were also isolated, though much less frequently (Figure 2). RAPD profiles allowed us to identify eight distinct genotyping profiles corresponding, after sequencing the isolates, to *D. hansenii* DH1, DH2, and DH3, and *D. tyrocola* DT1, *K. marxianus* KM1, *S. cerevisiae* SC1, *P. kudriavzevii* PK, and *T. delbrueckii* TD1. *D. hansenii* DH1, DH2, and DH3 prevailed in various batches of defective PV cheeses and in ripened cheeses produced during the two on-site inspections. *D. tyrocola* DT1 was isolated from a single defective cheese and from ripened cheeses produced on 6 February 2024. *K. marxianus* KM1 and *S. cerevisiae* SC1 were isolated in curd before or after curd maturation of cheeses produced on both on-site inspections, whereas *T. delbrueckii* TD1 was isolated only from single defective cheeses. Notably, *D. hansenii* DH1 and DH3 were found to be dominant in cheeses produced during both on-site inspections (Figure 3, green and red squares). *Saccharomyces cerevisiae* SC1 and *K. marxianus* KM1 were also isolated from the drained whey of the previous day’s production, used to prepare the starter used for the PV production on the second on-site inspection (6 February 2024; Figure 2). The constant presence of strains SC1, KM1, DH1, and DH3 in different defective cheeses sampled between April and August 2023 (Table 1, single PV cheeses), as well as in samples (and corresponding ripened cheeses) taken during the two on-site inspections carried out 5 months apart, would suggest the presence of yeast biofilms in the dairy environment or along the production line. The ability of dairy yeasts, such as *D. hansenii* and *K. marxianus*, to produce biofilms in dairy processing facilities has previously been reported [21,22].

**Figure 2 foods-15-00129-f002:**
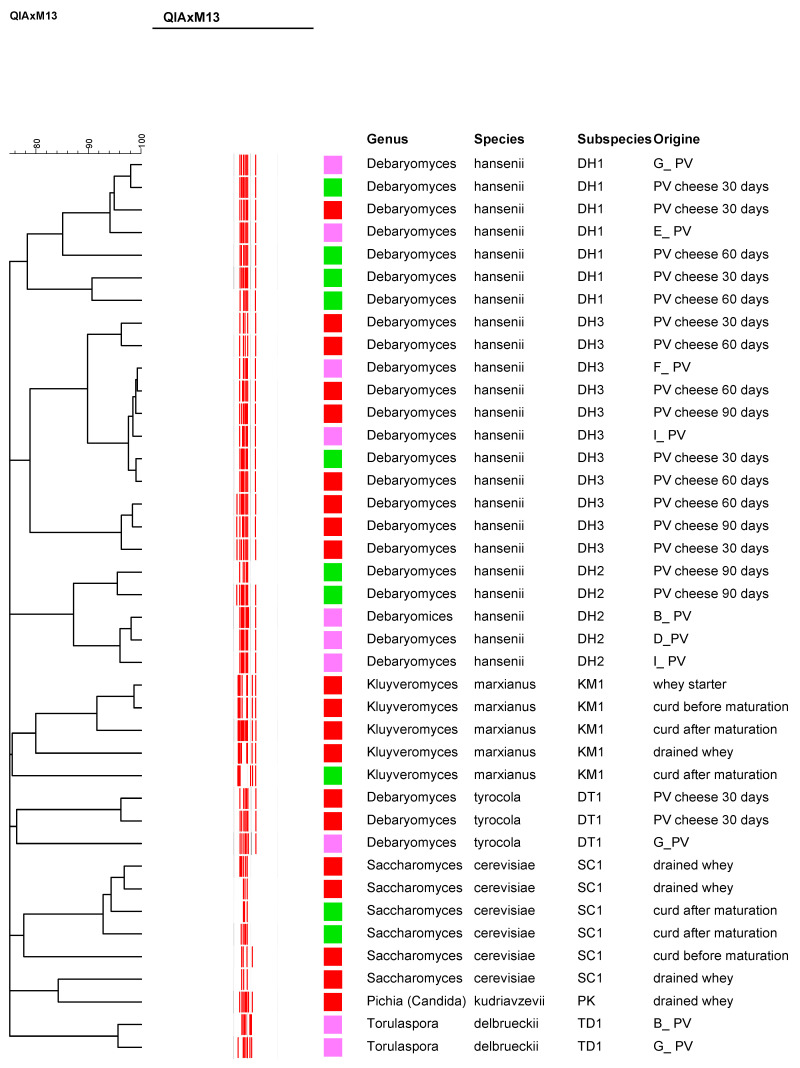
Cluster analysis of RAPD-PCR patterns obtained with primer M13 of 40 strains of yeasts isolated from single PV cheeses and along the production line during on-site inspections carried out on 21 September 2023 and 6 February 2024. Pearson’s correlation coefficient and the Ward’s method as a clustering algorithm were applied. On the right-side species, the origin of isolation and biotypes are indicated. Colored boxes are related to the origin of isolation: pink box, different defective cheese batches; green box, on-site inspection of 21 September 2023; red box, on-site inspection of 6 February 2024.

**Figure 3 foods-15-00129-f003:**
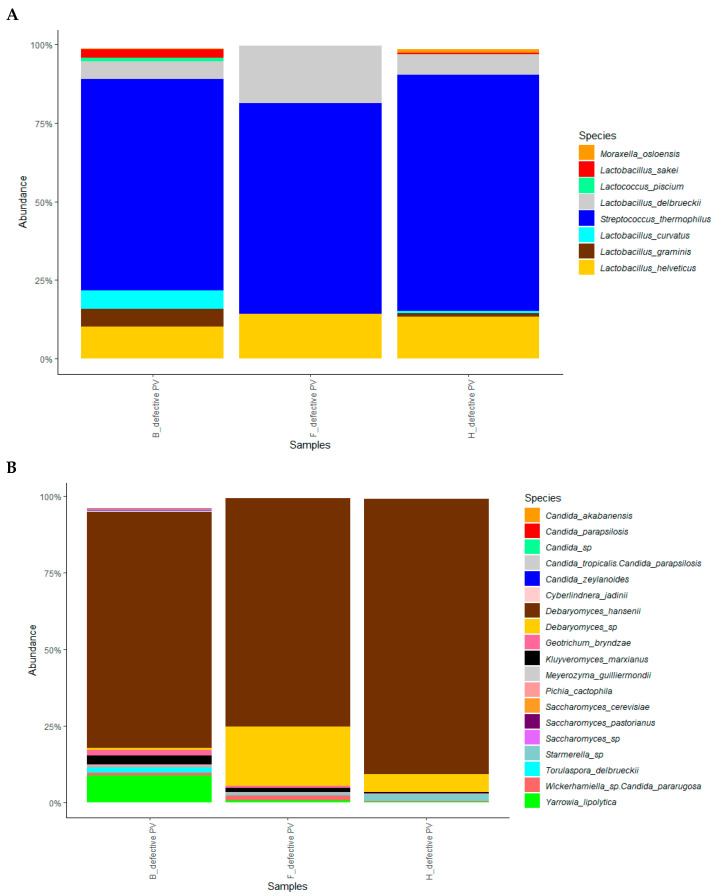
Metataxonomic analysis of three PV cheese samples with >log 4.0 CFU/mL of yeast count. Average values of relative abundance of (**A**) 8 bacterial dominant taxa and (**B**) 19 yeast dominant taxa were retrieved.

### 3.4. Metagenomic Analysis

Three single PV samples (B, F, and H defective PV) with a yeast count > log 4.0 CFU g^−1^ (Table 1) were analyzed by metabarcoding. Concerning 16S metabarcoding, 1,039,172 reads were sequenced, with 346,390 reads per sample on average (range of 307,075–372,594). Results showed the presence of eight dominant species, with a higher relative abundance of thermophilic LAB species such as *Lactobacillus delbrueckii*, *S. thermophilus*, and *Lactobacillus helveticus*, that, together, reached 83%, 99%, and 95% of the total dominant species in B, F, and H defective PV samples, respectively (Figure 3A). These species normally belong to the Provolone cheese microbiota as they are known to be prevalent in the natural starter cultures traditionally used in this technology [23]. No gas-forming bacterial taxa possibly suspected to have caused the defect were retrieved, thus confirming the results of the microbial counts. By yeast metabarcoding, 1,328,133 reads were sequenced, with 442,711 reads per sample on average (range of 377,770–542,975). Among 19 dominant yeast taxa, *Debaryomyces* spp. (especially *D. hansenii*) prevailed over the overall yeast community (about 78% in B and 94% in F and H defective PV samples) (Figure 3B). A lower presence of *S. cerevisiae*, *Yarrowia lipolytica*, and *K. marxianus* was also observed. Overall, metagenomic analysis corroborated microbiological data and molecular identification of yeast isolates (Figure 3), indicating yeasts (mainly belonging to *Debaryomyces* taxa) rather than bacteria as the causative agents of the defect.

### 3.5. Chemical Analysis

HPLC analysis of the curds and cheeses sampled during the on-site inspections showed a regular acidification, which lowered the pH to the expected value for stretching (4.94–5.01), with the relative increase in lactic acid after curd maturation (Table 2A,B). No other organic acids, such as propionic or butyric acids, due to possible fermentations carried out by propionibacteria or clostridia, were detected in both curds and cheeses. The consumption of sugars explains the different localization and dominance of yeasts highlighted above. During curd maturation, lactose was rapidly consumed, mainly due to NWC activity, while galactose accumulated in the curd before stretching. The rise in yeast population during curd maturation was probably due to the development of *K. marxianus* and *S. cerevisiae*, which are known to be able to ferment lactose and galactose [7,24,25], and therefore prevailed over other yeast species before ripening. *D. hansenii*, which assimilated residual galactose, tended instead to dominate in ripened cheeses. Spoiled cheeses contained a high quantity of alcohols in comparison with PV samples, as analyzed by Tidona et al. [26]. In this study, ethanol was detected in a very high amount, whereas the other alcohols were present in traces (Table 3). In the defective samples, a slight but significant rise in pH (Table 2; Figure 1), probably due to proteolysis and the reported ability of *D. hansenii* to assimilate lactic acid [27], was observed at the end of ripening.

The microbial agent(s) responsible for a structural defect (eyes) of sweet PV were identified. Considering the nature and characteristics of the defect, which consisted of widespread anomalies of the cheese paste during ripening due to gas accumulation, the investigation focused on the search and quantification of gas-producing microbial groups. Repeated sampling of cheeses affected by the defect and analyses of processing intermediates following two inspections at the cheese factory identified yeasts as the main causative agents. Yeasts are ubiquitous microorganisms in cheeses, where they, most often, can play a beneficial role but, due to the ability of some species to ferment lactose and/or galactose, can also be associated with spoilage [16,28,29]. Yeasts carry out alcoholic fermentation, with accumulation of ethyl alcohol and CO_2_ [7,24,25], which would explain the absence of taste irregularity and an anomalously high amount of ethyl alcohol in PV spoiled cheeses. The onset of paste microporosity generally occurred during ripening, a phase in which the yeast increased, while other potentially gas-forming microbial groups were in limited or negligible numbers.

On-site inspections highlighted a significant increase in yeasts, especially of two specific strains of *S. cerevisiae* and *K. marxianus*, in the maturing curd, followed by a substantial or partial ineffectiveness of the stretching in the reduction in yeasts, which survived at considerable levels, in the stretched curd. The survival of yeasts after curd stretching is not surprising, considering the increasing thermostability of these microorganisms [30,31]. Further development of yeasts took place up to 30 days of ripening, where *D. hansenii* dominated, giving rise to defective cheeses at 60 days. Chemical analysis highlighted that sugar consumption paralleled the microbial development, with *Kluyveromyces* and *Saccharomyces* (probably propagated with the NWC), which fermented lactose (*Kluyveromyces*) and galactose (*Kluyveromyces* and *Saccharomyces*), and therefore took over during curd maturation, followed by *Debaryomyces*, which fermented the residual galactose and overgrew in ripened cheeses. Genotypic typing suggested that specific yeast isolates seem to reside stably in the dairy, as confirmed by their repeated isolation from the same production line and from cheeses produced months apart.

Yeast contamination is a significant technical and economic concern in cheese production. Common sources of yeast contamination in dairy plants include raw materials, brine, air, wooden ripening shelves, and personnel. Therefore, maintaining a high hygiene standard and strict compliance with Good Hygiene Practices (GHP) and Good Manufacturing Practices (GMP) are essential prerequisites for limiting the number of yeasts and other harmful or undesirable microbiota in the production environment [16,20,29]. After the second on-site inspection, careful sanitization and hygiene of the PV production lines, equipment, and surrounding environment allowed a reduction in yeast numbers, especially along the critical points of the process (maturation and stretching of the curd; ), helping to restore the quality standard for this type of product. Similar conclusions were reached in a study in which yeasts were found to be spoilage agents in white brine cheeses, once again emphasizing the importance of proper hygiene and manufacturing practices during cheese processing to prevent and control yeast contamination [8].

## 4. Conclusions

The microbial agent causing a defect in the paste of sweet PV cheese, consisting of a widespread presence of holes, but without evidence of cheese blowing or taste deviations, was studied. To our knowledge, this is the first time that a defect with these characteristics has been described in PV cheese. A limited number of yeast strains belonging to species typically associated with cheese (*S. cerevisiae*, *K. marxianus*, and *D. hansenii*) found the conditions to colonize the cheese production line and developed further, especially during curd maturation and cheese ripening. This study demonstrated that yeast contamination along the production line could lead to quality defects in the PV. Strict compliance with GHP and GMP is recommended to limit the development of yeasts and other possible contaminants so as not to compromise cheese quality. Although yeasts are recognized as major spoilage agents, current understanding of species-specific spoilage mechanisms and metabolic pathways leading to gas, pigments, and off-flavors must be improved. Further research is needed to understand ecological and yeast–matrix interactions to develop predictive and intervention tools for large-scale industrial operations.

## Figures and Tables

**Table 2 foods-15-00129-t002:** Monitoring of pH, sugars, and organic acid during cheesemaking and ripening of Provolone Valpadana sampled on on-site inspections of 21 September 2023 (**A**) and 6 February 2024 (**B**). Values are expressed as g/100 g.

(**A**)									
**Samples**	**pH**	**Lactose**	**Glucose**	**Galactose**	**Lactic Acid**	**Acetic Acid**	**Succinic** **Acid**	**Propionic Acid**	**Butyric Acid**
curd before maturation	5.74	1.52	0.41	0.54	0.32	0.01	0.00	0.00	0.00
curd after maturation	5.01	0.05	0.15	0.61	1.05	0.02	0.01	0.00	0.00
curd after stretching	4.97	0.00	0.00	0.27	1.06	0.03	0.02	0.00	0.00
PV cheese 30 days	5.43	0.00	0.00	0.02	1.16	0.03	0.02	0.00	0.00
PV cheese 60 days	5.44	0.00	0.00	0.00	1.33	0.04	0.02	0.00	0.00
PV cheese 90 days	5.49	0.00	0.00	0.00	1.38	0.04	0.02	0.00	0.00
(**B**)									
**Samples**	**pH**	**Lactose**	**Glucose**	**Galactose**	**Lactic Acid**	**Acetic Acid**	**Succinic** **Acid**	**Propionic Acid**	**Butyric Acid**
curd before maturation	6.30	1.60	0.14	0.27	0.19	0.01	0.00	0.00	0.00
curd after maturation	4.94	0.06	0.03	0.37	1.32	0.04	0.03	0.00	0.00
curd after stretching	4.87	0.03	0.00	0.01	1.33	0.04	0.03	0.00	0.00
PV cheese 30 days	5.38	0.00	0.00	0.00	1.34	0.05	0.04	0.00	0.00
PV cheese 60 days	5.36	0.00	0.00	0.00	1.36	0.05	0.04	0.00	0.00
PV cheese 90 days	5.46	0.00	0.00	0.00	1.34	0.06	0.04	0.00	0.00

**Table 3 foods-15-00129-t003:** Volatile compounds detected in defective Provolone Valpadana cheeses. Values expressed as area units ×10^−6^.

	Mean	SD
2-propanol	23.2	±0.4
ethanol	298.0	±8.8
1-propanol	0.0	0.0
2-methyl-1-propanol	5.8	±0.3
2-pentanol	1.3	±0.1
1-butanol	12.6	±1.9
3-methyl-1-butanol	25.6	±0.4
3-buten-1-ol, 3-methyl	7.8	±0.8
2-buten-1-ol, 3-methyl	3.2	±0.5
2-butanol	3.4	±0.6
S of alcohols	381.1	±13.0

## Data Availability

The original contributions presented in this study are included in the article/Appendix A. Further inquiries can be directed to the corresponding author.

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
