# Peer review of "Detection, Identification, and Diffusion of Yeasts Responsible for Structural Defects in Provolone Valpadana PDO Cheese Using Multiple Research Techniques"

_foods, 2026, doi:10.3390/foods15010129_

Round 1
Reviewer 1 Report
Comments and Suggestions for Authors
The microbial agent causing a defect in the paste of sweet PV cheese, consisting of a widespread presence of holes, but without evidence of cheese blowing or taste deviations, was studied. This is the first time that a defect with these characteristics has been described in PV cheese. The study demonstrated that yeast contamination along the production line could lead to quality defects in the PV. This is an interesting study. There are several issues in this paper.
(1) line 58. studies on the microbiology of Provolone cheese are surprisingly limited. Microbial research in cheese should be common. Discussion and corresponding references need to be added.
(2) line 66. What is (1)?
(3) line 73. Microbial spoilage of cheeses is difficult to recognize and classify because it is generally caused by a variety of microorganisms and results from multiple and concomitant factors, linked to the quality of the raw material and the observance of process hygiene. Add sources and references for this sentence.
(4) line 172. Volatile alcohols were evaluated by SPME/GC/MS according to Lazzi et al. Need to add instrument model and parameters. How to achieve accurate qualitative analysis? Is there a standard product available?
(5) There is some overlap between Figure 1 and Figure 2, so Figure 1 can be removed. Or merge Figure 1 and Figure 2.
(6) two on-site inspections (23.09.21 and 24.02.06). Add corresponding introduction in 2.1 Cheese sampling. What's the difference between these two? Why choose these two times?
(7) Table 1, What is Sample label? A-I are?
(8) Table 3: Volatile compounds detected in defective Provolone Valpadana cheeses. How to achieve quantitative analysis? Internal standard method or standard curve?
(9) There should be a discussion that the study compared favourably with other references.
(10) Without the future trends. Add a section on existing research gaps and future trends.
Reviewer 2 Report
Comments and Suggestions for Authors
The authors identified a problem with the appearance of Provolone cheese due to gas-producing contaminating microorganisms and set out to identify the culprits. They took samples at all stages of the cheese-making process and carried out microorganism isolation, DNA extraction and genotyping. They identified S. cerevisiae and D. hansenii as the most likely causes of contamination and point out that these organisms can ferment lactose and galactose. In addition, they identify the stages in the production process where contaminating organisms increased in numbers and used chromatography to show which contaminating metabolites were being produced. The authors suggest that contamination might have been caused by the presence of biofilms somwhere in the production line. They further report that the problem was resolved by improved sanitization and hygiene.
This is a nice demonstration of the use of microbiological analysis to solve a problem in industry and so the article is of interest to microbiologists in both academic and industrial settings. I would have liked to know more about biofilms in the cheese plant but realise that it is not practical to close a busy plant in order to seek the source of contamination and that sterilization was a much more sensible solution. The figures and tables are informative and clear, as are the methods. The English is of a very high standard and I only spotted a few things that might be improved (see below). I recommend publication after minor corrections.
22: “Pichia kudriavsevii” should be “Pichia kudriavzevii”
43: should be a space before unit i.e. “52°C” should be “52 °C”
68: “accounting” needs another word after it (“for?”)
70: Delete”The” at the start of the sentence
100: word missing before “used”
106,107: add space before unit in temperature
107: plural so use “spp.” please
114: “Yeasts” should be “Yeast” (we say “cell culture” not “cells culture”)
120: space in temperature please
130: there should be no space before the percentage sign
154: Add space before “U” please
160: check unit as ng is a mass, not a concentration
171: words missing at start of sentence: does not make sense
Figure 1: better to use “5.38” rather than “5,38”
186: “eyes” should be “eye” (see earlier)
206: unit missing (“days?”)
Table 1: better to replace “,” with “.”
Table 3: again, it would be better to use “.” rather than “,”
328: “2” of carbon dioxide should be subscript (“CO2”)
332: replace “irrelevant” with “negligible”
336: “surviving” should be “survival”
349: define ”GHP and GMP”
